# Human Papilloma Virus Vaccination as a Strategy to Eliminate Cervical Cancer: Challenges and Opportunities

**DOI:** 10.3390/vaccines13030297

**Published:** 2025-03-11

**Authors:** Maria L. Avila-Aguero, Sebastian Ospina-Henao, Helena Brenes-Chacon, Carlos Espinal-Tejada, Ruby Trejo-Varon, Ana Morice

**Affiliations:** 1Instituto de Investigación en Ciencias Médicas (IICIMED), San José 10108, Costa Rica; 2Pediatric Infectious Diseases Division, Hospital Nacional de Niños “Dr. Carlos Sáenz Herrera”, San José 10103, Costa Rica; 3Center for Infectious Disease Modeling and Analysis (CIDMA), Yale University New Haven, New Haven, CT 06520, USA; 4Faculty of Medicine, Universidad de Ciencias Médicas, San José 10108, Costa Rica; 5Hospital Nacional de Niños “Dr. Carlos Sáenz Herrera”, Caja Costarricense de Seguro Social (CCSS), San José 10103, Costa Rica; helena.brenes@ucr.ac.cr; 6Global Health Consortium (GHC), Department of Global Health, Robert Stempel College of Public Health & Social Work, Florida International University, Miami, FL 33199, USA; caespina@fiu.edu (C.E.-T.); rtrejova@fiu.edu (R.T.-V.); 7International Independent Consultant, San José, Costa Rica; moriceana@gmail.com

**Keywords:** human papilloma virus, human papilloma virus vaccination, cervical cancer, Latin America

## Abstract

In August 2020, the World Health Assembly approved the global strategy to eliminate cervical cancer, envisioning a world where it seizes to be a public health problem. The cervical cancer elimination initiative reinforces the commitment to fulfilling the rights of adolescent girls and women by reducing both health and economic inequities in the poorest populations that have more limited access to timely and effective services. This initiative improves the quality of life of women and their families by protecting them from a disease that causes disability and preventing avoidable mortality through public health measures. This article discusses the epidemiological situation and vaccination coverage and identifies weaknesses and opportunities in Latin America and the Caribbean to propose actions to reinforce progress toward the cervical cancer elimination goal.

## 1. Introduction

In August 2020, the World Health Assembly approved the global strategy to eliminate cervical cancer, envisioning a world where it seizes to be a public health problem. Advances in knowledge, screening methods, timely detection, technological development, and innovation, such as human papillomavirus (HPV) vaccination, indicate that elimination could be achieved [1].

Although the implementation of cervical cancer prevention, detection, and treatment strategies has progressed globally and regionally, gaps that need to be addressed remain. The cervical cancer elimination initiative reinforces the commitment to respect the rights of adolescent girls and women by reducing both health and economic inequities [2] in settings with limited access to timely and effective services.

The COVID-19 pandemic affected the global progress of immunization programs and deepened the gaps in vaccination coverage [3]. The vaccination of school children against HPV was especially affected due to school closings and suspension of in-person classes as measures to mitigate the pandemic [4].

The aim of this document is to review the epidemiological situation of cervical cancer and HPV vaccination coverage, and to identify weaknesses and opportunities in Latin America and the Caribbean (LAC) to propose actions to reinforce progress toward the cervical cancer elimination goal.

Different data sources have been used to prepare this document: Documentary review: A bibliographic review was conducted in different databases: EBSCO, PubMed, and New England Journal of Medicine, using the keywords “HPV”, “Cervical Cancer”, “HPV Vaccine”, “Latin America”, resulting in 84 articles, of which 49 were used, which were chosen because they were relevant either to HPV vaccination in Latin America and the Caribbean or to cervical cancer screening and treatment.

Epidemiological analysis: of the incidence and mortality from cervical cancer related to economic income and human development variables. The registry of the International Association of Cancer Registries (IARC) and the World Health Organization (WHO) was used, which includes 185 countries and territories of the world updated to the year 2020. These data and visualizations are available online as public information on the site: https://gco.iarc.fr/today/about (accessed on 24 January 2022).

HPV vaccination: strategies and trends in vaccination coverage since the year of introduction of the countries of the region of the Americas were analyzed. Absolute data and administrative and official vaccination coverage against human papillomavirus (HPV) reported annually using the WHO/UNICEF Joint Immunization Reporting Form (JRF) and World Health Organization estimates were used. The effect of the SARS-CoV-2 pandemic on the level of coverage between LAC countries was evaluated by comparing trends from 2010 with respect to the first pandemic year 2020. https://immunizationdata.who.int/pages/coverage/hpv.html (accessed on 25 January 2022).

Socio-economic variables: cervical cancer incidence and mortality data and HPV vaccination coverage were analyzed with respect to the economic income categories of the countries reported by the World Bank (https://datatopics.worldbank.org/world-development-indicators/the-world-by-income-and-region.html (accessed on 25 January 2022)) and the Human Development Index using estimates prepared by the United Nations Development Program (UNDP) reported for the year 2020: http://hdr.undp.org/en/2020-report (accessed on 25 January 2022).

This paper is the product of a technical document from the Sociedad Latinoamericana de Infectología (SLIPE) prepared by the authors and available at https://slipe.org/web/wp-content/uploads/2022/05/Construyendo-la-agenda.pdf (accessed on 25 January 2022).

## 2. Global Situation

Cervical cancer is the fourth most common type of cancer among women globally, with around 660,000 new cases in 2022, and close to 94% of the 350,000 deaths caused by cervical cancer occurred in low- and middle-income countries. The highest rates of cervical cancer incidence and mortality are in sub-Saharan Africa (SSA), Central America, and South-East Asia [5,6]. Age-standardized incidence rates range from 23.8% in women from low-income countries (LICs) to 8.4% in those from high-income countries (HICs). Mortality rates follow the same trend, ranging from 2.5% to 17.4% in LICs and HICs, respectively [7]. Inequalities between regions are evident when considering cervical cancer incidence and mortality [8] according to the social development index (SDI). African countries have the lowest SDI and the highest cervical cancer incidence and mortality, while Europe and North America have a greater SDI and the lowest incidence and mortality rates [9].

In 2020, 84.6% of all cervical cancer cases and 91.4% of deaths related to this neoplasia were reported in low- and middle-income countries (LMICs). By this year, 41.3% of associated deaths occurred in LICs, compared with only 12.7% in HICs [7].

It is expected that the global burden of cervical cancer worsens, and by 2030, close to 700,000 new cases and approximately 400,000 deaths will occur due to cervical cancer [10], with a continue increase in discrepancies among LMICs and HICs.

## 3. Situation in Latin America and the Caribbean

Socioeconomic differences among countries in LAC have an impact on screening, diagnosis, and early treatment of cervical cancer and its previous stages [11]. They also play a role in the availability and coverage of HPV vaccination [12].

In 2022, more than 56,000 women were diagnosed with cervical cancer in LAC, with close to 28,000 related deaths [13]. Bolivia (10.7/100,000), Paraguay (11.5/100,000), and Venezuela (12/100,000) have the highest incidence of cervical cancer in LAC [14], while Uruguay 4.8/100,000), Costa Rica (4.5/100,000), Chile (6.3/100,000), and Puerto Rico (3.8/100,000) report the lowest rates [7]. Regarding mortality, Paraguay is the country with the highest rate, followed by Bolivia, Nicaragua, and Honduras. Conversely, Puerto Rico, Chile, Costa Rica, and Uruguay are the countries with the lowest mortality rates [7].

Furthermore, a comprehensive analysis of cervical cancer incidence and mortality in LAC countries underscores this troubling trend. It reveals that countries with a lower Human Development Index (HDI) exhibit a higher incidence of this disease. This correlation becomes even more pronounced when examining mortality rates, which are significantly elevated in nations with lower levels of development. This phenomenon can be attributed to various factors, such as limited access to quality healthcare services, a lack of early detection programs, and insufficient educational resources on cancer prevention. Therefore, it is crucial to implement public health policies that address these disparities and promote equitable access to medical care, particularly in the most affected regions. [15].

In 2009, the Food and Drug Administration licensed the quadrivalent human papillomavirus vaccine (HPV4; Gardasil, produced by Merck & Co., Inc., Rahway, NJ USA) [16]. That same year, the World Health Organization (WHO) published its first position paper with recommendations regarding age, gender, indication, dosing, and schedule [17]. The recommendations have changed over time, and countries are progressively expanding HPV vaccination programs to include men to provide direct protection, improve population-level prevention of HPV infection and/or disease, and encourage vaccine acceptance [18,19,20].

As of 2023, 140 of the 195 WHO Member States had introduced the HPV vaccine into national immunization programs [21]. Europe and the Americas have made the greatest progress in introducing the HPV vaccine into their national immunization schedules, with 77% and 85% of their countries, respectively [22]. During 2019, many LMICs made significant efforts to introduce this vaccine into their immunization programs, accelerating their progress toward the 2030 goal of “cervical cancer elimination”, a goal to be achieved with a coverage of at least 90% [23].

LAC countries have been successful in controlling vaccine-preventable infectious diseases, and the introduction of the HPV vaccine is no exception [24]. Countries that invest in preventive health include the HPV vaccine in their national immunization schedules, either partially or totally. Nevertheless, countries like Haiti, Nicaragua, Cuba, and Venezuela are among the countries in the region with no HPV vaccine among their national immunization programs [24] (Table 1), and also among the countries with current complex political, social, and economic issues in the region [25].

In LAC, Panama introduced the HPV vaccine in 2008 [26] and has already gained extensive experience, and progression to elimination targets is feasible in the region [27]. Nevertheless, the impact of vaccination will become evident probably after at least 10 years of introduction. Benign and premalignant lesions have been shown to decrease with the introduction of vaccination [28,29,30]. LAC is the region with the greatest progress in HPV vaccine implementation, but their monitoring systems are poor, and the available coverage data are scarce [31].

A comparative modeling analysis carried out in a recent study that included projections from three independent transmission-dynamic models found consistent results suggesting that HPV vaccination coverage of 90% of girls can lead to cervical cancer elimination in most LMICs [32]. Countries with higher incidences of cervical cancer (more than 25 cases per 100,000 women per year) may not obtain good results unless they accelerate not only vaccination but also screening tests [33]. Screening would accelerate cervical cancer elimination, and such elimination is necessary in countries with high incidence [33].

Intensive expansion of HPV vaccination in LMIC among girls is expected to halve the age-standardized incidence of cervical cancer by 2048. More than 74 million cervical cancer cases are expected to be prevented with both screening and vaccination, and 61 million with vaccination alone. It has been documented that HPV vaccination alone can lead to cervical cancer elimination globally, except in sub-Saharan Africa and LAC, where only 27% and 80%, respectively, will achieve this goal with no other preventive or screening measures such as sexual education, access to barrier planning methods such as condoms, gender-neutral vaccination and timely and equitable access to health services [32].

## 4. Vaccination Strategies and Coverage

VPH vaccination coverage from 2010 to 2020 in the Americas is shown in Figure 1. Even though an increase has been documented, it does not reach 80% of the target population [34]. In 2020, a significant decline in HPV vaccine coverage was observed, reflecting the impact of the COVID-19 pandemic on immunization programs [24] due to social mobility restrictions, suspension of in-person classes at schools, and diversion of economic and human resources, among others.

According to coverage data for the first and second doses of HPV vaccine in LAC country (Figure 2 and Figure 3), only Saint Lucia, Brazil, Saint Kitts and Nevis, Mexico, and Costa Rica had a coverage above 80% for the first dose [34].

An HPV vaccine coverage in the targeted population of at least 90% is required to achieve the goal of elimination [22]. An important indicator for monitoring coverage is the proportion of women with both doses at the age of fifteen. This percentage has progressively increased since 2011, with the previously mentioned decline experienced during the pandemic (Figure 2 and Figure 3) [34]. Nevertheless, the recent pandemic is not the only thing responsible for these statistics, as this behavior has been documented in some countries for some of the other five core vaccines (DPT, polio, and measles) when compared with coverage rates obtained in 2013-17 [35]. Studies have reported other causes of HPV vaccine hesitancy that could explain this change, such as low perception of benefits among parents, doubts about effectiveness, and fear of side effects, among others [36,37]. Unfortunate examples of vaccination campaigns have impacted the perception people may have of a proven successful prevention tool [38].

Strategic and concrete actions are required to improve these indicators. Appropriate strategies for HPV vaccine implementation, including patient–provider communication, education, integrated monitoring, and evaluation of used strategies, are necessary in LAC [39]. HPV vaccination programs are at risk of the same operative problems experienced by cervical cancer screening programs, an effective life-saving tool that is unfortunately underutilized for cancer prevention in LMIC [40].

The ongoing implementation of HPV vaccination in LAC requires surveillance [41]. The unique social and structural barriers related to HPV vaccination need to be addressed, and a robust and timely response should be established to make a significant impact on HPV-related cancers [42].

## 5. Strategic Actions to Achieve 90% HPV Vaccination Coverage

LAC has extensive experience in vaccination, achieving control and elimination goals for some vaccine-preventable diseases [43]. Vaccination is considered a cornerstone of public health, but immunization programs based in school settings (schools and universities) have collapsed to some extent due to the pandemic [44]. Identifying gaps caused by school dropouts and unvaccinated adolescents through an aggressive catch-up vaccination campaign represents part of the work to be conducted [45]. However, identifying missed opportunities for vaccination is not easy without national immunization records. This problem can be solved by involving civil societies in the identification of unvaccinated girls and educating and advocating for their vaccination as one of the cornerstones of successful immunization [46]. Demystifying HPV vaccination requires a solid communication campaign, addressing doubts, and educating families about HPV immunization [47].

In addition to high vaccination coverage, secondary prevention should be improved with adequate screening programs and timely treatment through equitable access to healthcare services [48].

## 6. Conclusions

The COVID-19 pandemic had a major impact on communities, disrupting many essential healthcare services [49], including the provision of routine immunization services, exposing the fragility of healthcare systems around the world. However, in the post-pandemic era, we must make a critical analysis of all those aspects that we can improve to get closer to the goal of eliminating HPV as a cause of uterine cancer.

Progress in HPV vaccination has been evident in Latin America through the years, placing this region alongside those with successful experiences, such as Australia or Europe.

Despite the progress in the introduction of the HPV vaccine, low coverage is a challenge faced by the LAC region, where communication and awareness campaigns are needed to improve immunization program success. Cervical cancer requires, in addition to prevention, early diagnosis, and timely treatment. Therefore, catch-up vaccination campaigns and community outreach programs for screening are necessary. Identification of disparities in HPV disease indicators related to the community’s living conditions and social determinants of health are necessary in the road to cervical cancer prevention. Governments and public–private partnerships should ensure access to economic resources and effective healthcare systems to improve and promote screening campaigns and primary prevention through vaccination.

## Figures and Tables

**Figure 1 vaccines-13-00297-f001:**
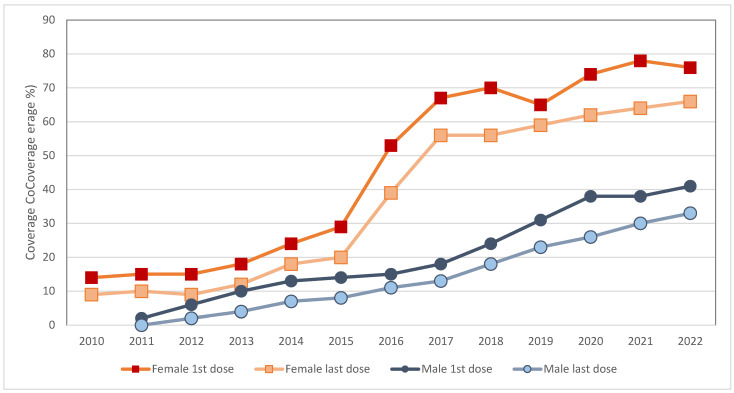
HPV vaccination coverage (first and last dose) by age 15 in female and male in the region of the Americas, 2010 a 2021. Source: Own elaboration based on data from WHO [34].

**Figure 2 vaccines-13-00297-f002:**
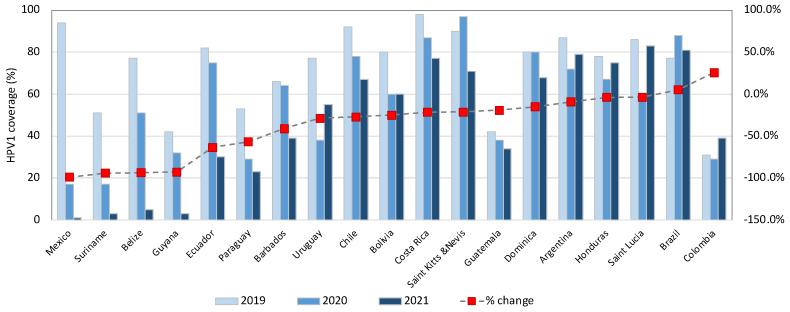
Change in HPV1 program coverage when comparing pre-pandemic year 2019 vs. 2021, Region of the Americas. Source. Own elaboration based on data from WHO [34].

**Figure 3 vaccines-13-00297-f003:**
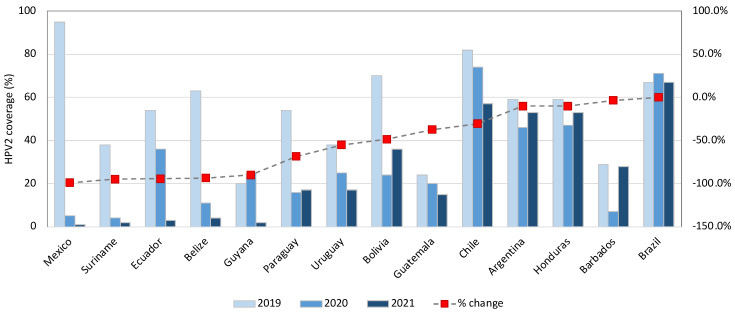
Change in HPV2 program coverage when comparing pre-pandemic year 2019 vs. 2021, region of the Americas. Source: Own elaboration based on data from WHO [34].

**Table 1 vaccines-13-00297-t001:** Year of HPV vaccine introduction (partial * or complete **) in the Americas, 2020.

Country	2006	2007	2008	2009	2010	2011	2012	2013	2014	2015	2016	2017	2018	2019	2020	Year
United States of America																2006
Puerto Rico																2007
Canada		Partial	Partial													2007
Bermuda																2007
Mexico			Partial	Partial	Partial	Partial										2008
Panama																2008
Argentina																2011
Guyana						Partial	Partial	Partial	Partial	Partial	Partial					2011
Colombia																2012
Trinidad and Tobago																2012
Brazil								Partial								2013
Paraguay																2013
Suriname																2013
Uruguay																2013
Barbados																2014
Ecuador																2014
Bahamas																2015
Chile																2015
Perú																2015
Belize																2016
Honduras																2016
Bolivia																2017
Jamaica																2017
Dominican Republic																2017
Saint Vincent and Grenadines																2017
Antigua y Barbuda																2018
Guatemala																2018
Costa Rica																2019
Dominica																2019
Grenada																2019
Saint Kitts and Nevis																2019
Saint Lucia																2019
El Salvador																2020
Cuba																No
Haiti																No
Nicaragua																No
Venezuela																No

* Does meet the standard of the universal immunization program/** meets the standard of the universal immunization program. Source: Own elaboration based on data from WHO [25].

## Data Availability

Data are contained in the article.

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
