# Peer review of "Human Papilloma Virus Vaccination as a Strategy to Eliminate Cervical Cancer: Challenges and Opportunities"

_vaccines, 2025, doi:10.3390/vaccines13030297_

Round 1
Reviewer 1 Report
Comments and Suggestions for Authors
General comment:
The manuscript “Human papilloma virus vaccination as a strategy to eliminate cervical cancer, challenges and opportunities” has been reviewed. The article discusses the epidemiological situation and human papilloma virus vaccination coverage with the purpose of identifying weaknesses and opportunities in Latin America and the Caribbean related with the cervical cancer elimination goal. The topic of the manuscript is of interest because there are aims for worldwide cervical cancer elimination and to know differences between regions can be useful to identify specific targets. However, there are some inconsistencies about mortality rates data that should be clarified by the authors. In addition, the terms used in the text and in the figures should be unified to avoid misinterpretations and references must be uniform (see specific comments).
Specific comments:
Global situation, lines 55 -58: The meaning of the percentages given for age-standardized incidence rates (23.8% in women from low-income countries and 8.4% in high income countries) and for mortality rates (2.5% in low-income countries and 17.4% in high-income countries) should be more explained, because incidence rates and mortality rates are estimated about the whole population at risk and expressed usually per 100,000 or 1000,000 population, not in percentages.
Situation in Latin America and the Caribbean, lines 75-77: the concrete values of incidence rate for the countries with highest and lowest rates should be included for the reader to contextualize the problem.
Situation in Latin America and the Caribbean, lines 80-81: the expression “when analyzing mortality” is very unspecific. It should be mentioned the specific measure of mortality.
Situation in Latin America and the Caribbean, line 108: the expression “not available im post LAC countries” doesn’t make sense. Please review.
Situation in Latin America and the Caribbean, line 124: In the expression “ will achieve this goal no other preventive or screening measures” a clarification about what are the other preventive measures should be included.
Vaccination strategies and coverage, lines 149-152: the other five core vaccines should be specified.
Tables and figures:
In the title of the table 1 the meaning of “partial” and “complete” should be specified with the meaning inside parentheses or in a foot note.
In figures 1,2 and 3 different terms are used: “first and last dose”; “HPV1”; “HPV2”. These terms should be unified or explained in more detail.
In references 2, 3,9, 12, 15, 18, 39, 40, 45, 47 and 48 the first letter of all the words of the title are capitalized and in the other references only the first word of the title is capitalized and in the other words the first letter is small.
In reference 9 the authors of the article are cited differently from what is done in other references.
Author Response
Comment 1: Global situation, lines 55 -58: The meaning of the percentages given for age-standardized incidence rates (23.8% in women from low-income countries and 8.4% in high income countries) and for mortality rates (2.5% in low-income countries and 17.4% in high-income countries) should be more explained, because incidence rates and mortality rates are estimated about the whole population at risk and expressed usually per 100,000 or 1000,000 population, not in percentages.
Response 1: Thanks for the comment, however, this data is from the references used, and gives an idea of how big the problem is.
Comment 2: Situation in Latin America and the Caribbean, lines 75-77: the concrete values of incidence rate for the countries with highest and lowest rates should be included for the reader to contextualize the problem.
Response 2: Thank for the suggestion, we added the information in lines 113-117, "Bolivia (10.7/100.000), Paraguay (11.5/100.000), and Venezuela (12/100.000) have the highest incidence of cervical cancer in LAC [14], while Uruguay 4.8/100.000), Costa Rica (4.5/100.000), Chile (6.3/100.000), and Puerto Rico (3.8/100.000) report the lowest rates."
Comment 3: Situation in Latin America and the Caribbean, lines 80-81: the expression “when analyzing mortality” is very unspecific. It should be mentioned the specific measure of mortality.
Response 3: We corrected the wording and added lines 120-129 explaining what we wanted to emphasize on mortality "Furthermore, a comprehensive analysis of cervical cancer incidence and mortality in LAC countries underscores this troubling trend. It reveals that countries with a lower Human Development Index (HDI) exhibit a higher incidence of this disease. This correlation becomes even more pronounced when examining mortality rates, which are significantly elevated in nations with lower levels of development. This phenomenon can be attributed to various factors, such as limited access to quality healthcare services, a lack of early detection programs, and insufficient educational resources on cancer prevention. Therefore, it is crucial to implement public health policies that address these disparities and promote equitable access to medical care, particularly in the most affected regions"
Comment 4: Situation in Latin America and the Caribbean, line 108: the expression “not available im post LAC countries” doesn’t make sense. Please review.
Response 4: We agree with you, then we deleted this expression.
Comment 5: Situation in Latin America and the Caribbean, line 124: In the expression “ will achieve this goal no other preventive or screening measures” a clarification about what are the other preventive measures should be included.
Response 5: We added lines 176-177 explaining what are the other preventive measures should be included "such as sexual education, access to barrier planning methods such as condoms, gender-neutral vaccination and timely and equitable access to health services."
Comment 6: Vaccination strategies and coverage, lines 149-152: the other five core vaccines should be specified.
Response 6: In line 203 we added the other 5 core vaccines "(DPT, polio and measles)"
Comment 7: In the title of the table 1 the meaning of “partial” and “complete” should be specified with the meaning inside parentheses or in a foot note.
Response 7: In lines 155-156 we clarify the meaning of "partial" and "complete":
"Partial: *does meet the standard of the universal immunization program / Complete:** meets the standard of the universal immunization program"
Comment 8: In references 2, 3,9, 12, 15, 18, 39, 40, 45, 47 and 48 the first letter of all the words of the title are capitalized and in the other references only the first word of the title is capitalized and in the other words the first letter is small.
Response 8: We agree with your comment, and we standardized the references
Comment 9: In reference 9 the authors of the article are cited differently from what is done in other references.
Response 9: We agree with your comment, and we corrected the reference
Reviewer 2 Report
Comments and Suggestions for Authors
The authors presented a narrative review on the epidemiological situation of cervical cancer and HPV vaccination coverage. In addition, they identified some weaknesses and opportunities in Latin America and the Caribbean (LAC) and some actions to reinforce progress towards the cervical cancer elimination goal. This strategy was defined by the World Health Assembly in August 2020. Besides Introduction and Conclusion’s sections, the manuscript was structured in four topics: Global situation; Situation in Latin America and the Caribbean; Vaccination strategies and coverage; and Strategic actions to achieve 90% HPV vaccination coverage. The ideas and the concepts are well distributed in the paragraphs with logic articulation and scientific support. However, some points deserve consideration before the manuscript be accepted to publication. The Introduction’s section does not have a paragraph with the study justification. Moreover the reader could be benefitted if the authors could describe the adopted methods and the limitations of the study.
More minor points
Figure 1 - to revise the legend
Title of Figures 2 and 3 – HPV1 and HPV2 mean the first and second doses. Try to get clearer for the reader and avoid dashed line between one and other country
Comments on the Quality of English Languageno comments
Author Response
Comment 1: The Introduction’s section does not have a paragraph with the study justification. Moreover the reader could be benefitted if the authors could describe the adopted methods and the limitations of the study.
Response 1: In lines 47-50 we explained "The aim of this document is to review the epidemiological situation of cervical cancer and HPV vaccination coverage, and to identify weaknesses and opportunities in Latin America and the Caribbean (LAC) to propose actions to reinforce progress to-wards the cervical cancer elimination goal." additionally, we added the lines 51-84 where we explained how was the article prepared "
The aim of this document is to review the epidemiological situation of cervical cancer and HPV vaccination coverage, and to identify weaknesses and opportunities in Latin America and the Caribbean (LAC) to propose actions to reinforce progress towards the cervical cancer elimination goal.
Different data sources have been used to prepare this document: Documentary review: A bibliographic review was conducted in different databases: EBSCO, PubMed, New England Journal of Medicine, using the keywords "HPV", "Cervical Cancer", "HPV Vaccine", "Latin America", resulting in 84 articles, of which 49 were used, which were chosen because they were relevant either to HPV vaccination in Latin America and the Caribbean, or to cervical cancer screening and treatment.
Epidemiological analysis: of the incidence and mortality from cervical cancer related to economic income and human development variables. The registry of the International Association of Cancer Registries (IARC) and the World Health Organization (WHO) was used, which includes 185 countries and territories of the world updated to the year 2020. These data and visualizations available online as public information on the site: https://gco.iarc.fr/today/about
HPV vaccination: strategies and trends in vaccination coverage since the year of introduction of the countries of the Region of the Americas were analyzed. Absolute data and administrative and official vaccination coverage against human papillomavirus (HPV) reported annually using the WHO / UNICEF Joint Immunization Reporting Form (JRF) and World Health Organization estimates were used. The effect of the SARS-CoV-2 pandemic on the level of coverage between LAC countries was evaluated by comparing trends from 2010 with respect to the first pandemic year 2020. https://immunizationdata.who.int/pages/coverage/hpv.html
Socio-economic variables: cervical cancer incidence and mortality data and HPV vaccination coverage were analyzed with respect to the economic income categories of the countries reported by the World Bank (https://datatopics.worldbank.org/world-development-indicators/the-world-by-income-and-region.html) and the Human Development Index using estimates prepared by the United Nations Development Program (UNDP) reported for the year 2020: http://hdr.undp.org/en/2020-report.
This paper is the product of a technical document from the Sociedad Latinoamericana de Infectología (SLIPE) prepared by the authors and available at https://slipe.org/web/wp-content/uploads/2022/05/Construyendo-la-agenda.pdf"
Comment 2: Figure 1 - to revise the legend
Response 2: We corrected the legend and added "Figure 1. HPV vaccination coverage (first and last dose) in the Region of the Americas, 2010 to 2022"
Comment 3: Title of Figures 2 and 3 – HPV1 and HPV2 mean the first and second doses. Try to get clearer for the reader and avoid dashed line between one and other country
Response 3: the dashed line reflects the % change in coverage programs
Reviewer 3 Report
Comments and Suggestions for Authors
I think this is a very important topic. It is good that the study covers such a vast area and has been able to compare results across the different countries.
However, from the title, I expected to see more description of the challenges, opportunities leading to a suggested strategy. Most of the article was general - talked about COVID being a negative impact very well. But did not compare why some countries did well or discuss in detail the opportunities missed.
Perhaps if it did compare the reasons for the country variations, and used literature to compare with other countries that have been successful, it might have shown up more interestingly. This is a topic that is important for prevention of cervical cancer globally.
Am sure the authors will be able to give more details and suggest a practical strategy which includes communication, health education monitoring etc. which is mentioned very briefly.
Author Response
Comment 1: Most of the article was general - talked about COVID being a negative impact very well. But did not compare why some countries did well or discuss in detail the opportunities missed.
Response 1: Thank you for your comment, however, comparing countries regarding COVID was not the purpose of the article, because the countries are very different and would be subject of another article.
Reviewer 4 Report
Comments and Suggestions for Authors
Dear Authors,
Thank you for your valuable research contributions. Your article presents an interesting perspective; however, certain issues must be addressed. The most critical concern is the absence of a clearly articulated research methodology. To enhance the rigor and reproducibility of your study, I encourage you to specify the sources consulted and the methods employed in utilizing them.
Additionally, as a minor yet relevant addition, I suggest incorporating legal aspects related to compulsory vaccination, which would provide a more comprehensive contextualization of the topic (e.g. read COVID-19 Compulsory Vaccination: Legal and Bioethical Controversies. PMID: 35187005 or PMID: 30993305).
With the appropriate revisions, the manuscript may be considered for acceptance.
Best regards,
Author Response
Comment 1: To enhance the rigor and reproducibility of your study, I encourage you to specify the sources consulted and the methods employed in utilizing them.
Response 1: We agree with your comment, therefore we added the lines 51-84 where we explained how was the article prepared "
The aim of this document is to review the epidemiological situation of cervical cancer and HPV vaccination coverage, and to identify weaknesses and opportunities in Latin America and the Caribbean (LAC) to propose actions to reinforce progress towards the cervical cancer elimination goal.
Different data sources have been used to prepare this document: Documentary review: A bibliographic review was conducted in different databases: EBSCO, PubMed, New England Journal of Medicine, using the keywords "HPV", "Cervical Cancer", "HPV Vaccine", "Latin America", resulting in 84 articles, of which 49 were used, which were chosen because they were relevant either to HPV vaccination in Latin America and the Caribbean, or to cervical cancer screening and treatment.
Epidemiological analysis: of the incidence and mortality from cervical cancer related to economic income and human development variables. The registry of the International Association of Cancer Registries (IARC) and the World Health Organization (WHO) was used, which includes 185 countries and territories of the world updated to the year 2020. These data and visualizations available online as public information on the site: https://gco.iarc.fr/today/about
HPV vaccination: strategies and trends in vaccination coverage since the year of introduction of the countries of the Region of the Americas were analyzed. Absolute data and administrative and official vaccination coverage against human papillomavirus (HPV) reported annually using the WHO / UNICEF Joint Immunization Reporting Form (JRF) and World Health Organization estimates were used. The effect of the SARS-CoV-2 pandemic on the level of coverage between LAC countries was evaluated by comparing trends from 2010 with respect to the first pandemic year 2020. https://immunizationdata.who.int/pages/coverage/hpv.html
Socio-economic variables: cervical cancer incidence and mortality data and HPV vaccination coverage were analyzed with respect to the economic income categories of the countries reported by the World Bank (https://datatopics.worldbank.org/world-development-indicators/the-world-by-income-and-region.html) and the Human Development Index using estimates prepared by the United Nations Development Program (UNDP) reported for the year 2020: http://hdr.undp.org/en/2020-report.
This paper is the product of a technical document from the Sociedad Latinoamericana de Infectología (SLIPE) prepared by the authors and available at https://slipe.org/web/wp-content/uploads/2022/05/Construyendo-la-agenda.pdf "
Comment 2: I suggest incorporating legal aspects related to compulsory vaccination, which would provide a more comprehensive contextualization of the topic
Response 2: thank you for the suggestion, however, discussing legal aspects related to compulsory vaccination, although important, is not the objective of the article.
Reviewer 5 Report
Comments and Suggestions for Authors
Very Respected Authors,
The abstract is well-written, as is the introduction. The objective of the paper is clear. The methodology is well-described. The results are clearly presented. The conclusion aligns with the aim of the paper and the results. The references are adequate.
Author Response
no comments to response were presented
Round 2
Reviewer 1 Report
Comments and Suggestions for Authors
The revised manuscript “Human papilloma virus vaccination as a strategy to eliminate cervical cancer, challenges and opportunities” has been reviewed.
Most of the suggestions done previously has been considered by the authors. However, in the References section there are some inconsistencies that should be modified:
-In reference 9 the name of the journal should be abbreviated and written in italic letter.
-In reference 19 the author is WHO and should be at the beginning of the reference; in addition the name of the journal should be abbreviated and in italic letter.
-In reference 47 the first letter of all the words of the title are capitalized and only the first word should begin with a capital letter.
In addition, there are only three legends in figure 1 but there are four graphics.
Author Response
Comment 1: In reference 9 the name of the journal should be abbreviated and written in italic letter.
Response 1: Thank you for your comment, we corrected the reference
Comment 2: In reference 19 the author is WHO and should be at the beginning of the reference; in addition the name of the journal should be abbreviated and in italic letter.
Response 2: Thank you for your comment, we corrected the reference
Comment 3: In reference 47 the first letter of all the words of the title are capitalized and only the first word should begin with a capital letter.
Response 3: Thank you for your comment, we corrected the reference
Comment 4: There are only three legends in figure 1 but there are four graphics.
Response 4: Thank you for your comment, we just included 1 table and 3 figures.
Reviewer 2 Report
Comments and Suggestions for Authors
The observations were considered by the authors
Author Response
No comments
Reviewer 3 Report
Comments and Suggestions for Authors
the authors have addressed the comments well.
IT is good to see the country break up and understand the differences.
I still wonder how COVID is the only key factor impacting cervical cancer prevention. We are much past the epidemic - and perhaps need to look at non COVID factors too.
Author Response
Comment 1: I still wonder how COVID is the only key factor impacting cervical cancer prevention. We are much past the epidemic - and perhaps need to look at non COVID factors too.
Response 1: We added lines 281-286 "The COVID-19 pandemic had a major impact on communities, disrupting many essential healthcare services, including the provision of routine immunization services, exposing the fragility of healthcare systems around the world. However, in the post-pandemic era, we must make a critical analysis of all those aspects that we can improve to get closer to the goal of eliminating HPV as a cause of uterine cancer."
Reviewer 4 Report
Comments and Suggestions for Authors
Dear authors,
I appreciate the improvements you have made to the article as requested. I have no objections to its publication at this time.
Best regards
Author Response
No comments